# Bayesian Optimization with Exponential Convergence

**Kenji Kawaguchi**
MIT
Cambridge, MA, 02139
kawaguch@mit.edu

**Leslie Pack Kaelbling**
MIT
Cambridge, MA, 02139
lpk@csail.mit.edu

**Tomás Lozano-Pérez**
MIT
Cambridge, MA, 02139
tlp@csail.mit.edu

## Abstract

This paper presents a Bayesian optimization method with exponential convergence *without* the need of auxiliary optimization and *without* the $\delta$-cover sampling. Most Bayesian optimization methods require auxiliary optimization: an additional non-convex global optimization problem, which can be time-consuming and hard to implement in practice. Also, the existing Bayesian optimization method with exponential convergence [1] requires access to the $\delta$-cover sampling, which was considered to be impractical [1, 2]. Our approach eliminates both requirements and achieves an exponential convergence rate.

## 1 Introduction

We consider a general global optimization problem: maximize $f(x)$ subject to $x \in \Omega \subset \mathbb{R}^D$ where $f : \Omega \to \mathbb{R}$ is a non-convex black-box deterministic function. Such a problem arises in many real-world applications, such as parameter tuning in machine learning [3], engineering design problems [4], and model parameter fitting in biology [5]. For this problem, one performance measure of an algorithm is the *simple regret*, $r_n$, which is given by $r_n = \sup_{x \in \Omega} f(x) - f(x^+)$ where $x^+$ is the best input vector found by the algorithm. For brevity, we use the term "regret" to mean simple regret.

The general global optimization problem is known to be intractable if we make no further assumptions [6]. The simplest additional assumption to restore tractability is to assume the existence of a bound on the slope of $f$. A well-known variant of this assumption is Lipschitz continuity with a known Lipschitz constant, and many algorithms have been proposed in this setting [7, 8, 9]. These algorithms successfully guaranteed certain bounds on the regret. However appealing from a theoretical point of view, a practical concern was soon raised regarding the assumption that a tight Lipschitz constant is known. Some researchers relaxed this somewhat strong assumption by proposing procedures to estimate a Lipschitz constant during the optimization process [10, 11, 12].

Bayesian optimization is an efficient way to relax this assumption of complete knowledge of the Lipschitz constant, and has become a well-recognized method for solving global optimization problems with non-convex black-box functions. In the machine learning community, Bayesian optimization—especially by means of a Gaussian process (GP)—is an active research area [13, 14, 15]. With the requirement of the access to the $\delta$-cover sampling procedure (it samples the function uniformly such that the density of samples doubles in the feasible regions at each iteration), de Freitas et al. [1] recently proposed a theoretical procedure that maintains an exponential convergence rate (exponential regret). However, as pointed out by Wang et al. [2], one remaining problem is to derive a GP-based optimization method with an exponential convergence rate *without* the $\delta$-cover sampling procedure, which is computationally too demanding in many cases.

In this paper, we propose a novel GP-based global optimization algorithm, which maintains an exponential convergence rate and converges rapidly *without* the $\delta$-cover sampling procedure.

## 2 Gaussian Process Optimization

In Gaussian process optimization, we estimate the distribution over function $f$ and use this information to decide which point of $f$ should be evaluated next. In a parametric approach, we consider a parameterized function $f(x; \theta)$, with $\theta$ being distributed according to some prior. In contrast, the non-parametric GP approach directly puts the GP prior over $f$ as $f(\cdot) \sim GP(m(\cdot), \kappa(\cdot, \cdot))$ where $m(\cdot)$ is the mean function and $\kappa(\cdot, \cdot)$ is the covariance function or the kernel. That is, $m(x) = \mathbb{E}[f(x)]$ and $\kappa(x, x') = \mathbb{E}[(f(x) - m(x))(f(x') - m(x'))^T]$. For a finite set of points, the GP model is simply a joint Gaussian: $\mathbf{f}(\mathbf{x}_{1:N}) \sim \mathcal{N}(\mathbf{m}(\mathbf{x}_{1:N}), \mathbf{K})$, where $\mathbf{K}_{i,j} = \kappa(x_i, x_j)$ and $N$ is the number of data points. To predict the value of $f$ at a new data point, we first consider the joint distribution over $f$ of the old data points and the new data point:

$$\begin{pmatrix} \mathbf{f}(\mathbf{x}_{1:N}) \\ f(x_{N+1}) \end{pmatrix} \sim \mathcal{N}\left( \begin{pmatrix} \mathbf{m}(\mathbf{x}_{1:N}) \\ m(x_{N+1}) \end{pmatrix}, \begin{bmatrix} \mathbf{K} & \mathbf{k} \\ \mathbf{k}^T & \kappa(x_{N+1}, x_{N+1}) \end{bmatrix} \right)$$

where $\mathbf{k} = \kappa(\mathbf{x}_{1:N}, \mathbf{x}_{N+1}) \in \mathbb{R}^{N \times 1}$. Then, after factorizing the joint distribution using the Schur complement for the joint Gaussian, we obtain the conditional distribution, conditioned on observed entities $\mathcal{D}_N := \{\mathbf{x}_{1:N}, \mathbf{f}(\mathbf{x}_{1:N})\}$ and $x_{N+1}$, as:

$$f(\mathbf{x}_{N+1}) | \mathcal{D}_N, x_{N+1} \sim \mathcal{N}(\mu(x_{N+1}|\mathcal{D}_N), \sigma^2(x_{N+1}|\mathcal{D}_N))$$

where $\mu(x_{N+1}|\mathcal{D}_N) = m(x_{N+1}) + \mathbf{k}^T\mathbf{K}^{-1}(\mathbf{f}(\mathbf{x}_{1:N}) - \mathbf{m}(\mathbf{x}_{1:N}))$ and $\sigma^2(x_{N+1}|\mathcal{D}_N) = \kappa(\mathbf{x}_{N+1}, \mathbf{x}_{N+1}) - \mathbf{k}^T\mathbf{K}^{-1}\mathbf{k}$. One advantage of GP is that this closed-form solution simplifies both its analysis and implementation.

To use a GP, we must specify the mean function and the covariance function. The mean function is usually set to be zero. With this zero mean function, the conditional mean $\mu(x_{N+1}|\mathcal{D}_N)$ can still be flexibly specified by the covariance function, as shown in the above equation for $\mu$. For the covariance function, there are several common choices, including the Matern kernel and the Gaussian kernel. For example, the Gaussian kernel is defined as $\kappa(x, x') = \exp\left(-\frac{1}{2}(x - x')^T\Sigma^{-1}(x - x')\right)$ where $\Sigma^{-1}$ is the kernel parameter matrix. The kernel parameters or hyperparameters can be estimated by empirical Bayesian methods [16]; see [17] for more information about GP.

The flexibility and simplicity of the GP prior make it a common choice for continuous objective functions in the Bayesian optimization literature. Bayesian optimization with GP selects the next query point that optimizes the acquisition function generated by GP. Commonly used acquisition functions include the upper confidence bound (UCB) and expected improvement (EI). For brevity, we consider Bayesian optimization with UCB, which works as follows. At each iteration, the UCB function $\mathcal{U}$ is maintained as $\mathcal{U}(x|D_N) = \mu(x|\mathcal{D}_N) + \varsigma\sigma(x|\mathcal{D}_N)$ where $\varsigma \in \mathbb{R}$ is a parameter of the algorithm. To find the next query $x_{n+1}$ for the objective function $f$, GP-UCB solves an additional non-convex optimization problem with $\mathcal{U}$ as $x_{N+1} = \arg\max_x \mathcal{U}(x|D_N)$. This is often carried out by other global optimization methods such as DIRECT and CMA-ES. The justification for introducing a new optimization problem lies in the assumption that the cost of evaluating the objective function $f$ dominates that of solving additional optimization problem.

For deterministic function, de Freitas et al. [1] recently presented a theoretical procedure that maintains exponential convergence rate. However, their own paper and the follow-up research [1, 2] point out that this result relies on an impractical sampling procedure, the $\delta$-cover sampling. To overcome this issue, Wang et al. [2] combined GP-UCB with a hierarchical partitioning optimization method, the SOO algorithm [18], providing a regret bound with polynomial dependence on the number of function evaluations. They concluded that creating a GP-based algorithm with an *exponential* convergence rate *without* the impractical sampling procedure remained an open problem.

## 3 Infinite-Metric GP Optimization

### 3.1 Overview

The GP-UCB algorithm can be seen as a member of the class of bound-based search methods, which includes Lipschitz optimization, A* search, and PAC-MDP algorithms with optimism in the face of uncertainty. Bound-based search methods have a common property: the tightness of the bound determines its effectiveness. The tighter the bound is, the better the performance becomes.

However, it is often difficult to obtain a tight bound while maintaining correctness. For example, in A* search, admissible heuristics maintain the correctness of the bound, but the estimated bound with admissibility is often too loose in practice, resulting in a long period of global search.

The GP-UCB algorithm has the same problem. The bound in GP-UCB is represented by UCB, which has the following property: $f(x) \leq \mathcal{U}(x|\mathcal{D})$ with some probability. We formalize this property in the analysis of our algorithm. The problem is essentially due to the difficulty of obtaining a tight bound $\mathcal{U}(x|\mathcal{D})$ such that $f(x) \leq \mathcal{U}(x|\mathcal{D})$ and $f(x) \approx \mathcal{U}(x|\mathcal{D})$ (with some probability). Our solution strategy is to first admit that the bound encoded in GP prior may not be tight enough to be useful by itself. Instead of relying on a single bound given by the GP, we leverage the existence of an *unknown* bound encoded in the continuity at a global optimizer.

**Assumption 1**. (Unknown Bound) There exists a global optimizer $x^*$ and an *unknown* semi-metric $\ell$ such that for all $x \in \Omega$, $f(x^*) \leq f(x) + \ell(x, x^*)$ and $\ell(x, x^*) < \infty$.

In other words, we do not expect the *known* upper bound due to GP to be tight, but instead expect that there exists some *unknown* bound that might be tighter. Notice that in the case where the bound by GP is as tight as the unknown bound by semi-metric $\ell$ in Assumption 1, our method still maintains an exponential convergence rate and an advantage over GP-UCB (no need for auxiliary optimization). Our method is expected to become relatively much better when the *known* bound due to GP is less tight compared to the unknown bound by $\ell$.

As the semi-metric $\ell$ is unknown, there are infinitely many possible candidates that we can think of for $\ell$. Accordingly, we simultaneously conduct global and local searches based on all the candidates of the bounds. The bound estimated by GP is used to reduce the number of candidates. Since the bound estimated by GP is known, we can ignore the candidates of the bounds that are looser than the bound estimated by GP. The source code of the proposed algorithm is publicly available at http://lis.csail.mit.edu/code/imgpo.html.

### 3.2 Description of Algorithm

Figure 1 illustrates how the algorithm works with a simple 1-dimensional objective function. We employ hierarchical partitioning to maintain hyperintervals, as illustrated by the line segments in the figure. We consider a hyperrectangle as our hyperinterval, with its center being the evaluation point of $f$ (blue points in each line segment in Figure 1). For each iteration $t$, the algorithm performs the following procedure *for each interval size*:

(i) Select the interval with the maximum center value among the intervals of the same size.

(ii) Keep the interval selected by (i) if it has a center value greater than that of any *larger* interval.

(iii) Keep the interval accepted by (ii) if it contains a UCB greater than the center value of any *smaller* interval.

(iv) If an interval is accepted by (iii), divide it along with the longest coordinate into three new intervals.

(v) For each new interval, if the UCB of the evaluation point is less than the best function value found so far, skip the evaluation and use the UCB value as the center value until the interval is accepted in step (ii) on some future iteration; otherwise, evaluate the center value.

(vi) Repeat steps (i)–(v) until every size of intervals are considered

Then, at the end of each iteration, the algorithm updates the GP hyperparameters. Here, the purpose of steps (i)–(iii) is to select an interval that might contain the global optimizer. Steps (i) and (ii) select the possible intervals based on the unknown bound by $\ell$, while Step (iii) does so based on the bound by GP.

We now explain the procedure using the example in Figure 1. Let $n$ be the number of divisions of intervals and let $N$ be the number of function evaluations. $t$ is the number of iterations. Initially, there is only one interval (the center of the input region $\Omega \subset \mathbb{R}$) and thus this interval is divided, resulting in the first diagram of Figure 1. At the beginning of iteration $t = 2$, step (i) selects the third interval from the left side in the first diagram ($t = 1, n = 2$), as its center value is the maximum. Because there are no intervals of different size at this point, steps (ii) and (iii) are skipped. Step (iv) divides the third interval, and then the GP hyperparameters are updated, resulting in the second

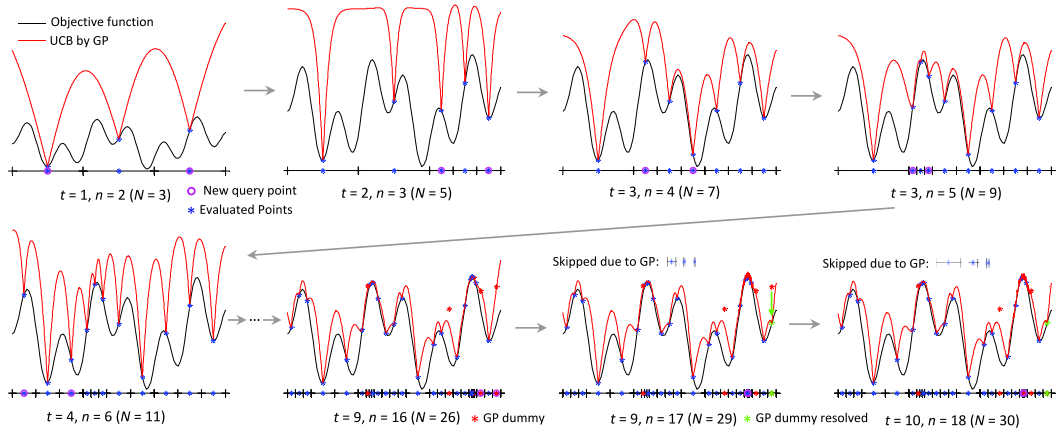

Figure 1: An illustration of IMGPO: $t$ is the number of iteration, $n$ is the number of divisions (or splits), $N$ is the number of function evaluations.

diagram ($t = 2, n = 3$). At the beginning of iteration $t = 3$, it starts conducting steps (i)–(v) for the largest intervals. Step (i) selects the second interval from the left side and step (ii) is skipped. Step (iii) accepts the second interval, because the UCB within this interval is no less than the center value of the smaller intervals, resulting in the third diagram ($t = 3, n = 4$). Iteration $t = 3$ continues by conducting steps (i)–(v) for the smaller intervals. Step (i) selects the second interval from the left side, step (ii) accepts it, and step (iii) is skipped, resulting in the forth diagram ($t = 3, n = 4$). The effect of the step (v) can be seen in the diagrams for iteration $t = 9$. At $n = 16$, the far right interval is divided, but no function evaluation occurs. Instead, UCB values given by GP are placed in the new intervals indicated by the red asterisks. One of the temporary dummy values is resolved at $n = 17$ when the interval is queried for division, as shown by the green asterisk. The effect of step (iii) for the rejection case is illustrated in the last diagram for iteration $t = 10$. At $n = 18$, $t$ is increased to 10 from 9, meaning that the largest intervals are first considered for division. However, the three largest intervals are all rejected in step (iii), resulting in the division of a very small interval near the global optimum at $n = 18$.

### 3.3 Technical Detail of Algorithm

We define $h$ to be the depth of the hierarchical partitioning tree, and $c_{h,i}$ to be the center point of the $i^{th}$ hyperrectangle at depth $h$. $N_{gp}$ is the number of the GP evaluations. Define $depth(\mathcal{T})$ to be the largest integer $h$ such that the set $\mathcal{T}_h$ is not empty. To compute UCB $\mathcal{U}$, we use $\varsigma_M = \sqrt{2\log(\pi^2 M^2/12\eta)}$ where $M$ is the number of the calls made so far for $\mathcal{U}$ (i.e., each time we use $\mathcal{U}$, we increment $M$ by one). This particular form of $\varsigma_M$ is to maintain the property of $f(x) \leq \mathcal{U}(x|\mathcal{D})$ during an execution of our algorithm with probability at least $1 - \eta$. Here, $\eta$ is the parameter of IMGPO. $\Xi_{max}$ is another parameter, but it is only used to limit the possibly long computation of step (iii) (in the worst case, step (iii) computes UCBs $3^{\Xi_{max}}$ times although it would rarely happen).

The pseudocode is shown in Algorithm 1. Lines 8 to 23 correspond to steps (i)-(iii). These lines compute the index $i_h^*$ of the candidate of the rectangle that may contain a global optimizer for each depth $h$. For each depth $h$, non-null index $i_h^*$ at Line 24 indicates the remaining candidate of a rectangle that we want to divide. Lines 24 to 33 correspond to steps (iv)-(v) where the remaining candidates of the rectangles for all $h$ are divided. To provide a simple executable division scheme (line 29), we assume $\Omega$ to be a hyperrectangle (see the last paragraph of section 4 for a general case).

Lines 8 to 17 correspond to steps (i)-(ii). Specifically, line 10 implements step (i) where a single candidate is selected for each depth, and lines 11 to 12 conduct step (ii) where some candidates are screened out. Lines 13 to 17 resolve the the temporary dummy values computed by GP. Lines 18 to 23 correspond to step (iii) where the candidates are further screened out. At line 21, $\mathcal{T}'_{h+\xi}(c_{h,i_h^*})$ indicates the set of *all* center points of a fully expanded tree until depth $h + \xi$ *within* the region covered by the hyperrectangle centered at $c_{h,i_h^*}$. In other words, $\mathcal{T}'_{h+\xi}(c_{h,i_h^*})$ contains the nodes of the fully expanded tree rooted at $c_{h,i_h^*}$ with depth $\xi$ and can be computed by dividing the current rectangle at $c_{h,i_h^*}$ and recursively divide all the resulting new rectangles until depth $\xi$ (i.e., depth $\xi$ from $c_{h,i_h^*}$, which is depth $h + \xi$ in the whole tree).

---
**Algorithm 1** Infinite-Metric GP Optimization (IMGPO)
---
**Input:** an objective function $f$, the search domain $\Omega$, the GP kernel $\kappa$, $\Xi_{max} \in \mathbb{N}^+$ and $\eta \in (0,1)$

1: Initialize the set $\mathcal{T}_h = \{\emptyset\} \ \forall h \geq 0$
2: Set $c_{0,0}$ to be the center point of $\Omega$ and $\mathcal{T}_0 \leftarrow \{c_{0,0}\}$
3: Evaluate $f$ at $c_{0,0}$: $g(c_{0,0}) \leftarrow f(c_{0,0})$
4: $f^+ \leftarrow g(c_{0,0}), \mathcal{D} \leftarrow \ \{(c_{0,0}, g(c_{0,0}))\}$
5: $n, N \leftarrow 1, N_{gp} \leftarrow 0, \Xi \leftarrow 1$
6: **for** $t = 1, 2, 3, ...$ **do**
7: $\quad v_{max} \leftarrow -\infty$
8: $\quad$ **for** $h = 0$ to $depth(\mathcal{T})$ **do** $\qquad\qquad\qquad\qquad\qquad\qquad$ # **for-loop for steps (i)-(ii)**
9: $\qquad$ **while** true **do**
10: $\qquad\quad i_h^* \leftarrow \ \arg\max_{i:c_{h,i}\in\mathcal{T}_h} g(c_{h,i})$
11: $\qquad\quad$ **if** $g(c_{h,i_h^*}) < v_{max}$ **then**
12: $\qquad\qquad i_h^* \leftarrow \emptyset$, **break**
13: $\qquad\quad$ **else if** $g(c_{h,i_h^*})$ is *not* labeled as *GP-based* **then**
14: $\qquad\qquad v_{max} \leftarrow g(c_{h,i_h^*})$, **break**
15: $\qquad\quad$ **else**
16: $\qquad\qquad g(c_{h,i_h^*}) \leftarrow f(c_{h,i_h^*})$ and remove the *GP-based* label from $g(c_{h,i_h^*})$
17: $\qquad\qquad N \leftarrow N+1, \ N_{gp} \leftarrow N_{gp}-1$
18: $\qquad\qquad \mathcal{D} \leftarrow \{\mathcal{D}, (c_{h,i_h^*}, g(c_{h,i_h^*}))\}$
19: $\quad$ **for** $h = 0$ to $depth(\mathcal{T})$ **do** $\qquad\qquad\qquad\qquad\qquad\qquad\quad$ # **for-loop for step (iii)**
20: $\qquad$ **if** $i_h^* \neq \emptyset$ **then**
21: $\qquad\quad \xi \leftarrow$ the smallest positive integer s.t. $i_{h+\xi}^* \neq \emptyset$ and $\xi \leq \min(\Xi, \Xi_{max})$ if exists, and 0 otherwise
22: $\qquad\quad z(h, i_h^*) = \max_{k:c_{h+\xi,k}\in\mathcal{T}'_{h+\xi}(c_{h,i_h^*})} \mathcal{U}(c_{h+\xi,k}|\mathcal{D})$
23: $\qquad\quad$ **if** $\xi \neq 0$ and $z(h, i_h^*) < g(c_{h+\xi, i_{h+\xi}^*})$ **then**
24: $\qquad\qquad i_h^* \leftarrow \emptyset$, **break**
25: $\quad v_{max} \leftarrow -\infty$
26: $\quad$ **for** $h = 0$ to $depth(\mathcal{T})$ **do** $\qquad\qquad\qquad\qquad\qquad\qquad$ # **for-loop for steps (iv)-(v)**
27: $\qquad$ **if** $i_h^* \neq \emptyset$ and $g(c_{h,i_h^*}) \geq v_{max}$ **then**
28: $\qquad\quad n \leftarrow n+1.$
29: $\qquad\quad$ Divide the hyperrectangle centered at $c_{h,i_h^*}$ along with the longest coordinate into three new hyperrectangles with the following centers:
$\qquad\qquad \mathcal{S} = \{c_{h+1,i(left)}, c_{h+1,i(center)}, c_{h+1,i(right)}\}$
30: $\qquad\quad \mathcal{T}_{h+1} \leftarrow \{\mathcal{T}_{h+1}, \mathcal{S}\}$
31: $\qquad\quad \mathcal{T}_h \leftarrow \mathcal{T}_h \setminus c_{h,i_h^*}, g(c_{h+1,i(center)}) \leftarrow g(c_{h,i_h^*})$
32: $\qquad\quad$ **for** $i_{new} = \{i(left), i(right)\}$ **do**
33: $\qquad\qquad$ **if** $\mathcal{U}(c_{h+1,i_{new}}|\mathcal{D}) \geq f^+$ **then**
34: $\qquad\qquad\quad g(c_{h+1,i_{new}}) \leftarrow f(c_{h+1,i_{new}})$
35: $\qquad\qquad\quad \mathcal{D} \leftarrow \{\mathcal{D}, (c_{h+1,i_{new}}, g(c_{h+1,i_{new}}))\}$
$\qquad\qquad\quad N \leftarrow N+1, f^+ \leftarrow \max(f^+, g(c_{h+1,i_{new}})), v_{max} = \max(v_{max}, g(c_{h+1,i_{new}}))$
36: $\qquad\qquad$ **else**
37: $\qquad\qquad\quad g(c_{h+1,i_{new}}) \leftarrow \mathcal{U}(c_{h+1,i_{new}}|\mathcal{D})$ and label $g(c_{h+1,i_{new}})$ as *GP-based*.
$\qquad\qquad\quad N_{gp} \leftarrow N_{gp}+1$
38: $\quad$ Update $\Xi$: if $f^+$ was updated, $\Xi \leftarrow \Xi + 2^2$ , and otherwise, $\Xi \leftarrow \max(\Xi - 2^{-1}, 1)$
39: $\quad$ Update GP hyperparameters by an empirical Bayesian method
---

## 3.4 Relationship to Previous Algorithms

The most closely related algorithm is the BaMSOO algorithm [2], which combines SOO with GP-UCB. However, it only achieves a polynomial regret bound while IMGPO achieves a exponential regret bound. IMGPO can achieve exponential regret because it utilizes the information encoded in the GP prior/posterior to reduce the degree of the unknownness of the semi-metric $\ell$.

The idea of considering a set of infinitely many bounds was first proposed by Jones et al. [19]. Their DIRECT algorithm has been successfully applied to real-world problems [4, 5], but it only maintains the consistency property (i.e., convergence in the limit) from a theoretical viewpoint. DIRECT takes an input parameter $\epsilon$ to balance the global and local search efforts. This idea was generalized to the case of an unknown semi-metric and strengthened with a theoretical support (finite regret bound) by

Munos [18] in the SOO algorithm. By limiting the depth of the search tree with a parameter $h_{max}$, the SOO algorithm achieves a finite regret bound that depends on *the near-optimality dimension*.

## 4  Analysis

In this section, we prove an exponential convergence rate of IMGPO and theoretically discuss the reason why the novel idea underling IMGPO is beneficial. The proofs are provided in the supplementary material. To examine the effect of considering infinitely many possible candidates of the bounds, we introduce the following term.

**Definition 1**. (Infinite-metric exploration loss). The infinite-metric exploration loss $\rho_t$ is the number of intervals to be divided during iteration $t$.

The infinite-metric exploration loss $\rho_\tau$ can be computed as $\rho_t = \sum_{h=1}^{depth(\mathcal{T})} \mathbb{1}(i_h^* \neq \emptyset)$ at line 25. It is the cost (in terms of the number of function evaluations) incurred by not committing to any particular upper bound. If we were to rely on a specific bound, $\rho_\tau$ would be minimized to 1. For example, the DOO algorithm [18] has $\rho_t = 1 \; \forall t \geq 1$. *Even if we know a particular upper bound*, relying on this knowledge and thus minimizing $\rho_\tau$ is not a good option *unless the known bound is tight enough compared to the unknown bound leveraged in our algorithm*. This will be clarified in our analysis. Let $\bar{\rho}_t$ be the maximum of the averages of $\rho_{1:t'}$ for $t' = 1, 2, ..., t$ (i.e., $\bar{\rho}_t \equiv \max(\{\frac{1}{t'} \sum_{\tau=1}^{t'} \rho_\tau \; ; \; t' = 1, 2, ..., t\})$.

**Assumption 2.** There exist $L > 0$, $\alpha > 0$ and $p \geq 1$ in $\mathbb{R}$ such that for all $x, x' \in \Omega$, $\ell(x', x) \leq L\|x' - x\|_p^\alpha$.

In Theorem 1, we show that the exponential convergence rate $O\left(\lambda^{N+N_{gp}}\right)$ with $\lambda < 1$ is achieved. We define $\Xi_n \leq \Xi_{max}$ to be the largest $\xi$ used so far with $n$ total node expansions. For simplicity, we assume that $\Omega$ is a square, which we satisfied in our experiments by scaling original $\Omega$.

**Theorem 1.** Assume Assumptions 1 and 2. Let $\beta = \sup_{x,x' \in \Omega} \frac{1}{2}\|x - x'\|_\infty$. Let $\lambda = 3^{-\frac{\alpha}{2CD\bar{\rho}_t}} < 1$. Then, with probability at least $1 - \eta$, the regret of IMGPO is bounded as

$$r_N \leq L(3\beta D^{1/p})^\alpha \exp\left(-\alpha \left[\frac{N + N_{gp}}{2CD\bar{\rho}_t} - \Xi_n - 2\right] \ln 3\right) = O\left(\lambda^{N+N_{gp}}\right).$$

Importantly, our bound holds for the best values of the unknown $L, \alpha$ and $p$ even though these values are not given. The closest result in previous work is that of BaMSOO [2], which obtained $\tilde{O}(n^{-\frac{2\alpha}{D(4-\alpha)}})$ with probability $1 - \eta$ for $\alpha = \{1, 2\}$. As can be seen, we have improved the regret bound. Additionally, in our analysis, we can see how $L$, $p$, and $\alpha$ affect the bound, allowing us to view the inherent difficulty of an objective function in a theoretical perspective. Here, $C$ is a constant in $N$ and is used in previous work [18, 2]. For example, if we conduct $2^D$ or $3^D - 1$ function evaluations per node-expansion and if $p = \infty$, we have that $C = 1$.

We note that $\lambda$ can get close to one as input dimension $D$ increases, which suggests that there is a remaining challenge in scalability for higher dimensionality. One strategy for addressing this problem would be to leverage additional assumptions such as those in [14, 20].

**Remark 1.** (The effect of the tightness of UCB by GP) If UCB computed by GP is "useful" such that $N/\bar{\rho}_t = \Omega(N)$, then our regret bound becomes $O\left(\exp\left(-\frac{N+N_{gp}}{2CD}\alpha \ln 3\right)\right)$. If the bound due to UCB by GP is too loose (and thus useless), $\bar{\rho}_t$ can increase up to $O(N/t)$ (due to $\bar{\rho}_t \leq \sum_{i=1}^t i/t \leq O(N/t)$), resulting in the regret bound of $O\left(\exp\left(-\frac{t(1+N_{gp}/N)}{2CD}\alpha \ln 3\right)\right)$, which can be bounded by $O\left(\exp\left(-\frac{N+N_{gp}}{2CD}\max(\frac{1}{\sqrt{N}}, \frac{t}{N})\alpha \ln 3\right)\right)$[1]. This is still better than the known results.

**Remark 2.** (The effect of GP) Without the use of GP, our regret bound would be as follows: $r_N \leq L(3\beta D^{1/p})^\alpha \exp(-\alpha[\frac{N}{2CD}\frac{1}{\tilde{\rho}_t} - 2] \ln 3)$, where $\bar{\rho}_t \leq \tilde{\rho}_t$ is the infinite-metric exploration loss without

GP. Therefore, the use of GP reduces the regret bound by increasing $N_{gp}$ and decreasing $\bar{\rho}_t$, but may potentially increase the bound by increasing $\Xi_n \leq \Xi$.

**Remark 3.** (The effect of infinite-metric optimization) To understand the effect of considering all the possible upper bounds, we consider the case without GP. If we consider all the possible bounds, we have the regret bound $L(3\beta D^{1/p})^\alpha \exp(-\alpha[\frac{N}{2CD}\frac{1}{\bar{\rho}_t} - 2]\ln 3)$ for *the best unknown L, $\alpha$ and $p$.* For standard optimization with a estimated bound, we have $L'(3\beta D^{1/p'})^{\alpha'} \exp(-\alpha'[\frac{N}{2C'D} - 2]\ln 3)$ for an estimated $L', \alpha'$, and $p'$. By algebraic manipulation, considering all the possible bounds has a better regret when $\tilde{\rho}_t^{-1} \geq \frac{2CD}{N\ln 3^\alpha}((\frac{N}{2C'D} - 2)\ln 3^{\alpha'} + 2\ln 3^\alpha - \ln \frac{L'(3\beta D^{1/p'})^{\alpha'}}{L(3\beta D^{1/p})^\alpha})$. For an intuitive insight, we can simplify the above by assuming $\alpha' = \alpha$ and $C' = C$ as $\tilde{\rho}_t^{-1} \geq 1 - \frac{Cc_2 D}{N}\ln \frac{L'D^{\alpha/p'}}{LD^{\alpha/p}}$. Because $L$ and $p$ are the ones that achieve the lowest bound, the logarithm on the right-hand side is always non-negative. Hence, $\tilde{\rho}_t = 1$ always satisfies the condition. When $L'$ and $p'$ are not tight enough, the logarithmic term increases in magnitude, allowing $\tilde{\rho}_t$ to increase. For example, if the second term on the right-hand side has a magnitude of greater than 0.5, then $\tilde{\rho}_t = 2$ satisfies the inequality. Therefore, even if we know the upper bound of the function, we can see that it may be better not to rely on this, but rather take the infinite many possibilities into account.

One may improve the algorithm with different division procedures than one presented in Algorithm 1. Accordingly, in the supplementary material, we derive an abstract version of the regret bound for IMGPO with a family of division procedures that satisfy some assumptions. This information could be used to design a new division procedure.

## 5 Experiments

In this section, we compare the IMGPO algorithm with the SOO, BaMSOO, GP-PI and GP-EI algorithms [18, 2, 3]. In previous work, BaMSOO and GP-UCB were tested with a pair of a handpicked good kernel and hyperparameters for each function [2]. In our experiments, we assume that the knowledge of good kernel and hyperparameters is unavailable, which is usually the case in practice. Thus, for IMGPO, BaMSOO, GP-PI and GP-EI, we simply used one of the most popular kernels, the isotropic Matern kernel with $\nu = 5/2$. This is given by $\kappa(x,x') = g(\sqrt{5||x-x'||^2/l})$, where $g(z) = \sigma^2(1 + z + z^2/3)\exp(-z)$. Then, we blindly initialized the hyperparameters to $\sigma = 1$

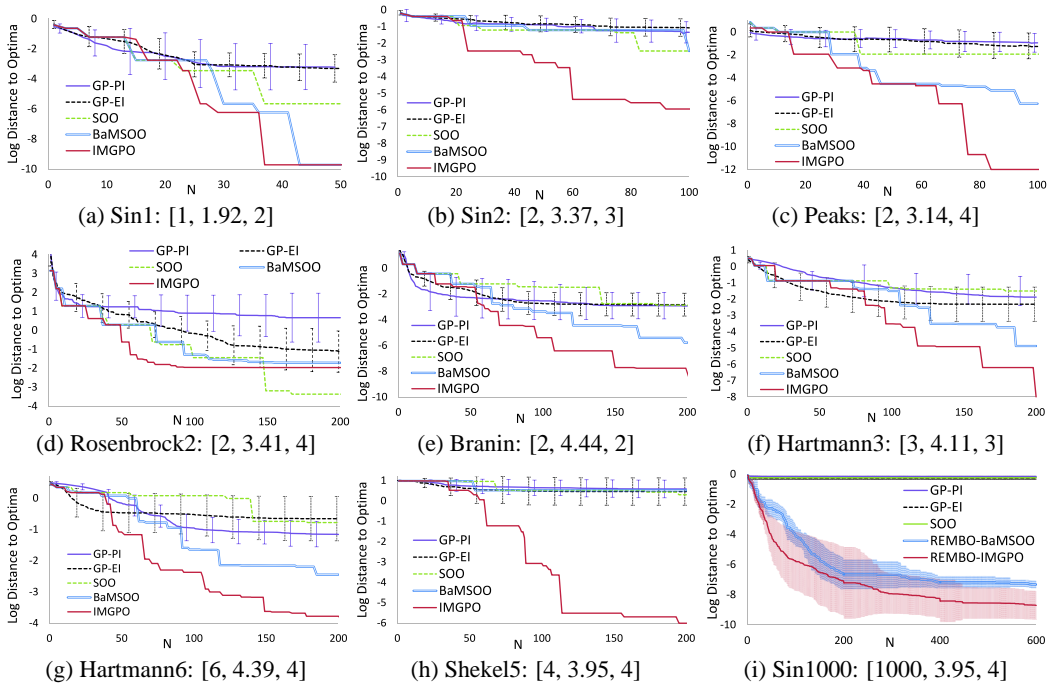

Figure 2: Performance Comparison: in the order, the digits inside of the parentheses [ ] indicate the dimensionality of each function, and the variables $\bar{\rho}_t$ and $\Xi_n$ at the end of computation for IMGPO.

Table 1: Average CPU time (in seconds) for the experiment with each test function

| Algorithm | Sin1 | Sin2 | Peaks | Rosenbrock2 | Branin | Hartmann3 | Hartmann6 | Shekel5 |
|---|---|---|---|---|---|---|---|---|
| GP-PI | 29.66 | 115.90 | 47.90 | 921.82 | 1124.21 | 573.67 | 657.36 | 611.01 |
| GP-EI | 12.74 | 115.79 | 44.94 | 893.04 | 1153.49 | 562.08 | 604.93 | 558.58 |
| SOO | 0.19 | 0.19 | 0.24 | 0.744 | 0.33 | 0.30 | 0.25 | 0.29 |
| BaMSOO | 43.80 | 4.61 | 7.83 | 12.09 | 14.86 | 14.14 | 26.68 | 371.36 |
| IMGPO | 1.61 | 3.15 | 4.70 | 11.11 | 5.73 | 6.80 | 13.47 | 15.92 |

and $l = 0.25$ for all the experiments; these values were updated with an empirical Bayesian method after each iteration. To compute the UCB by GP, we used $\eta = 0.05$ for IMGPO and BaMSOO. For IMGPO, $\Xi_{max}$ was fixed to be $2^2$ (the effect of selecting different values is discussed later). For BaMSOO and SOO, the parameter $h_{max}$ was set to $\sqrt{n}$, according to Corollary 4.3 in [18]. For GP-PI and GP-EI, we used the SOO algorithm and a local optimization method using gradients to solve the auxiliary optimization. For SOO, BaMSOO and IMGPO, we used the corresponding deterministic division procedure (given $\Omega$, the initial point is fixed and no randomness exists). For GP-PI and GP-EI, we randomly initialized the first evaluation point and report the mean and one standard deviation for 50 runs.

The experimental results for eight different objective functions are shown in Figure 2. The vertical axis is $\log_{10}(f(x^*) - f(x^+))$, where $f(x^*)$ is the global optima and $f(x^+)$ is the best value found by the algorithm. Hence, the lower the plotted value on the vertical axis, the better the algorithm's performance. The last five functions are standard benchmarks for global optimization [21]. The first two were used in [18] to test SOO, and can be written as $f_{sin1}(x) = (\sin(13x)\sin +1)/2$ for Sin1 and $f_{sin2}(x) = f_{sin1}(x_1)f_{sin1}(x_2)$ for Sin2. The form of the third function is given in Equation (16) and Figure 2 in [22]. The last function is Sin2 embedded in 1000 dimension in the same manner described in Section 4.1 in [14], which is used here to illustrate a possibility of using IMGPO as a main subroutine to scale up to higher dimensions with additional assumptions. For this function, we used REMBO [14] with IMGPO and BaMSOO as its Bayesian optimization subroutine. All of these functions are multimodal, except for Rosenbrock2, with dimensionality from 1 to 1000.

As we can see from Figure 2, IMGPO outperformed the other algorithms in general. SOO produced the competitive results for Rosenbrock2 because our GP prior was misleading (i.e., it did not model the objective function well and thus the property $f(x) \leq \mathcal{U}(x|\mathcal{D})$ did not hold many times). As can be seen in Table 1, IMGPO is much faster than traditional GP optimization methods although it is slower than SOO. For Sin 1, Sin2, Branin and Hartmann3, increasing $\Xi_{max}$ does not affect IMGPO because $\Xi_n$ did not reach $\Xi_{max} = 2^2$ (Figure 2). For the rest of the test functions, we would be able to improve the performance of IMGPO by increasing $\Xi_{max}$ at the cost of extra CPU time.

## 6 Conclusion

We have presented the first GP-based optimization method with an exponential convergence rate $O\left(\lambda^{N+N_{gp}}\right)$ ($\lambda < 1$) *without* the need of auxiliary optimization and the $\delta$-cover sampling. Perhaps more importantly in the viewpoint of a broader global optimization community, we have provided a practically oriented analysis framework, enabling us to see why *not* relying on a particular bound is advantageous, and how a non-tight bound can still be useful (in Remarks 1, 2 and 3). Following the advent of the DIRECT algorithm, the literature diverged along two paths, one with a particular bound and one without. GP-UCB can be categorized into the former. Our approach illustrates the benefits of combining these two paths.

As stated in Section 3.1, our solution idea was to use a bound-based method but rely less on the estimated bound by considering all the possible bounds. It would be interesting to see if a similar principle can be applicable to other types of bound-based methods such as planning algorithms (e.g., A* search and the UCT or FSSS algorithm [23]) and learning algorithms (e.g., PAC-MDP algorithms [24]).

**Acknowledgments**
The authors would like to thank Dr. Remi Munos for his thoughtful comments and suggestions. We gratefully acknowledge support from NSF grant 1420927, from ONR grant N00014-14-1-0486, and from ARO grant W911NF1410433. Kenji Kawaguchi was supported in part by the Funai Overseas Scholarship. Any opinions, findings, and conclusions or recommendations expressed in this material are those of the authors and do not necessarily reflect the views of our sponsors.

## Footnotes

[1]This can be done by limiting the depth of search tree as $depth(T) = O(\sqrt{N})$. Our proof works with this additional mechanism, but results in the regret bound with $N$ being replaced by $\sqrt{N}$. Thus, if we assume to have at least "not useless" UCBs such that $N/\bar{\rho}_t = \Omega(\sqrt{N})$, this additional mechanism can be disadvantageous. Accordingly, we do not adopt it in our experiments.

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
