[Supplementary Material · kawaguchi-nips15_supplementary.pdf]

# Bayesian Optimization with Exponential Convergence: Supplementary Material

**Kenji Kawaguchi**
MIT
Cambridge, MA, 02139
kawaguch@mit.edu

**Leslie Pack Kaelbling**
MIT
Cambridge, MA, 02139
lpk@csail.mit.edu

**Tomás Lozano-Pérez**
MIT
Cambridge, MA, 02139
tlp@csail.mit.edu

In this supplementary material, we provide the proofs of the theoretical results. Along the way, we also prove regret bounds for a general class of algorithms, the result of which may be used to design a new algorithm.

We first provide a known property of the upper confidence bound of GP.

**Lemma 1**. (Bound Estimated by GP) According to the belief encoded in the GP prior/posterior[1], for any $x$, $f(x) \leq \mathcal{U}(x|\mathcal{D})$ holds during the execution of Algorithm 1 with probability at least $1 - \eta$.

Proof. It follows the proof of lemma 5.1 of [1]. From the property of the standard gaussian distribution, $\Pr(f(x) > \mathcal{U}(x|\mathcal{D})) < \frac{1}{2}e^{-\varsigma_M^2/2}$. Taking union bound on the entire execution of Algorithm 1, $\Pr(f(x) > \mathcal{U}(x|\mathcal{D}) \ \forall M \geq 1) < \frac{1}{2}\sum_{M=1}^{\infty} e^{-\varsigma_M^2/2}$. Substituting $\varsigma_M = \sqrt{2\log(\pi^2 M^2/12\eta)}$, we obtain the statement. $\square$

Our algorithm has a concrete division procedure in line 27 of Algorithm 1. However, one may improve the algorithm with different division procedures. Accordingly, we first derive abstract version of regret bound for the IMGPO (Algorithm 1) under a family of division procedures that satisfy Assumptions 3 and 4. After that, we provide a proof for the main results in the paper.

## A   With Family of Division Procedure

In this section, we modify the result obtained by [2]. Let $x_{h,i}$ to be any point in the region covered by the $i^{th}$ hyperinterval at depth $h$, and $x_{h,i}^*$ be the global optimizer that may exist in the $i^{th}$ hyperinterval at depth $h$. The previous work provided the regret bound of the SOO algorithm with a family of division procedure that satisfies the following two assumptions.

**Assumption 3.** (Decreasing diameter) There exists a diameter function $\delta(h) > 0$ such that, for any hyperinterval $\omega_{h,i} \subset \Omega$ and its center $c_{h,i} \in \omega_{h,i}$ and any $x_{h,i} \in \omega_{h,i}$, we have $\delta(h) \geq \sup_{x_{h,i}} \ell(x_{h,i}, c_{h,i})$ and $\delta(h-1) \geq \delta(h)$ for all $h \geq 1$.

**Assumption 4.** (Well-shaped cell) There exists $\nu > 0$ such that any hyperinterval $\omega_{h,i}$ contains at least an $\ell$-ball of radius $\nu\delta(h)$ centered in $\omega_{h,i}$.

Thus, in this section, hyperinterval is not restricted to hyperrectangle. We now revisit the definitions of several terms and variables used in [2]. Let the $\epsilon$-optimal space $X_\epsilon$ be defined as $X_\epsilon := \{x \in \Omega : f(x) + \epsilon \geq f(x^*)\}$. That is, the $\epsilon$-optimal space is the set of input vectors whose function value is at least $\epsilon$-close to the global optima. To bound the number of hyperintervals relevant to this $\epsilon$-optimal space, we define a near-optimality dimension as follows.

**Definition 3.** (Near-optimality dimension) The near-optimality dimension is the smallest $d > 0$ such that, there exists $C > 0$, for all $\epsilon > 0$, the maximum number of disjoint $\ell$-balls of radius $\nu\epsilon$ with center in the $\epsilon$-optimal space $X_\epsilon$ is less than $C\epsilon^{-d}$.

Finally, we define the set of $\delta$-optimal hyperintervals $I_{\delta(h)}$ as $I_{\delta(h)} := \{\omega_{h,i} \ni c_{h,i} : f(c_{h,i}) + \delta(h) \geq f(x^*)\}$. The $\delta$-optimal hyperinterval $I_{\delta(h)}$ is used to relate the hyperintervals to the $\epsilon$-optimal space. Indeed, the $\delta$-optimal hyperinterval $I_{\delta(h)}$ is almost identical to the $\delta(h)$-optimal space $X_{\delta(h)}$, except that $I_{\delta(h)}$ is focused on the center points whereas $X_{\delta(h)}$ considers the whole input vector space. In the following, we use $|I_{\delta(h)}|$ to denote the number of $I_{\delta(h)}$ and derive its upper bound.

**Lemma 2.** (Lemma 3.1 in [2]) Let $d$ be the near-optimality dimension and $C$ denote the corresponding constant in Definition 1. Then, the number of $\delta$-optimal hyperintervals is bounded by $|I_{\delta(h)}| \leq C\delta(h)^{-d}$.

We are now ready to present the main result in this section. In the following, we use the term *optimal hyperinterval* to indicate a hyperinterval that contains a global optimizer $x^*$. We say a hyperinterval is *dominated* by other intervals when it is rejected or not selected in step (i)-(iii). In Lemma 3, we bound the maximum size of the optimal hyperinterval. From Assumption 1, this can be translated to the regret bound, as we shall see in Theorem 2.

**Lemma 3.** Let $\Xi_n \leq \min(\Xi, \Xi_{max})$ be the largest $\xi$ used so far with $n$ total node expansions. Let $h_n^*$ be the depth of the deepest expanded node that contains a global optimizer $x^*$ after $n$ total node expansions (i.e., $h_n^* \leq n$ determines the size of the *optimal hyperinterval*). Then, with probability at least $1 - \eta$, $h_n^*$ is bounded below by some $h'$ that satisfies

$$n \geq \sum_{\tau=1}^{\sum_{l=0}^{h'+\Xi}|I_l|} \rho_\tau.$$

*Proof.* Let $T_h$ denote the time at which the optimal hyperinterval is further divided. We prove the statement by showing that the time difference $T_{h+1} - T_h$ is bounded by the number of $\delta$-optimal hyperintervals. To do so, we first note that there are three types of hyperinterval that can dominate an optimal hyperinterval $c_{h+1,*}$ during the time $[T_h, T_{h+1} - 1]$, all of which belong to $\delta$-optimal hyperintervals $I_\delta$. The first type has the same size (i.e., same depth $h$), $c_{h+1,i}$. In this case,

$$f(c_{h+1,i}) \geq f(c_{h+1,*}) \geq f(x_{h+1,*}^*) - \delta(h+1),$$

where the first inequality is due to line 10 (step (i)) and the second follows Assumptions 1 and 2. Thus, it must be $c_{h+1,i} \in I_{h+1}$. The second case is where the optimal hyperinterval may be dominated by a hyperinterval of larger size (depth $l < h+1$), $c_{l,i}$. In this case, similarly,

$$f(c_{l,i}) \geq f(c_{h+1,*}) \geq f(x_{h+1,*}^*) - \delta(l),$$

where the first inequality is due to lines 11 to 12 (step (ii)) and thus $c_{l,i} \in I_l$. In the final scenario, the optimal hyperinterval is dominated by a hyperinterval of smaller size (depth $h+1+\xi$), $c_{h+1+\xi,i}$. In this case,

$$f(c_{h+1+\xi,i}) \geq z(h+1,*) \geq f(x_{h+1,*}^*) - \delta(h+1+\xi)$$

with probability at least $1 - \eta$ where $z(\cdot, \cdot)$ is defined in line 21 of Algorithm 1. The first inequality is due to lines 19 to 23 (step (iii)) and the second inequality follows Lemma 1 and Assumptions 1 and 3. Hence, we can see that $c_{h+1+\xi,i} \in I_{h+1+\xi}$.

For all of the above arguments, the temporarily assigned $\mathcal{U}$ under GP has no effect. This is because the algorithm still covers the above three types of $\delta$-optimal hyperintervals $I_\delta$, as $\mathcal{U} \geq f$ with probability at least $1 - \eta$ (Lemma 1). However, these are only expanded based on $f$ because of the temporary nature of $\mathcal{U}$. Putting these results together,

$$T_{h+1} - T_h \leq \sum_{\tau=1}^{\sum_{l=1}^{h+1+\Xi_n}|I_{\delta(l)}|} \rho_\tau.$$

Since if one of the $I_\delta$ is divided during $[T_h, T_{h+1} - 1]$, it cannot be divided again during another time period,

$$\sum_{h=0}^{h_n^*} T_{h+1} - T_h \leq \sum_{\tau=1}^{\sum_{l=1}^{h_n^*+1+\Xi_n}|I_l|} \rho_\tau,$$

where on the right-hand side, we could combine the summation $\sum_{h=0}^{h_n^*}$ and $\sum_{\tau=1}^{\sum_{l=1}^{h+1+\Xi_n}|I_{\delta(l)}|}$ into the one, because each $h$ in the summation refers to the same $\delta$-optimal interval $I_{\delta(l)}$ with $l \leq h_n^*+1+\Xi_n$, and should not be double-counted. As $\sum_{h=0}^{h_n^*} T_{h+1} - T_h = T_{h_n^*+1} - T_0$, $T_0 = 1$ and $|I_{\delta(0)}| = 1$,

$$T_{h_n^*+1} \leq 1 + \sum_{\tau=1}^{\sum_{l=1}^{h_n^*+1+\Xi_n}|I_l|} \rho_\tau \leq \sum_{\tau=1}^{\sum_{l=0}^{h_n^*+1+\Xi_n}|I_l|} \rho_\tau.$$

As $T_{h_n^*+1} > n$ by definition, for any $h'$ such that $\sum_{\tau=1}^{\sum_{l=0}^{h'+\Xi_n}|I_l|} \rho_\tau \leq n < \sum_{\tau=1}^{\sum_{l=0}^{h_n^*+1+\Xi_n}|I_l|} \rho_\tau$, we have $h_n^* > h'$. $\qquad \square$

With Lemmas 2 and 3, we are ready to present a finite regret bound with the family of division procedures.

**Theorem 2.** Assume Assumptions 1, 3, and 4. Let $h(n)$ be the smallest integer $h$ such that

$$n \leq \sum_{\tau=1}^{C\sum_{l=0}^{h+\Xi_n}\delta(l)^{-d}} \rho_\tau.$$

Then, with probability at least $1 - \eta$, the regret of the IMGPO with any general division procedure is bounded as

$$r_n \leq \delta(h(n) - 1).$$

*Proof.* Let $c(n)$ and $c_{h_n^*,*}$ be the center point expanded at the $n$th expansion and the optimal hyperinterval containing a global optimizer $x^*$, respectively. Then, from Assumptions 1, 3, and 4, $f(c(n)) \geq f(c_{h_n^*,*}) \geq f^* - \delta(h_n^*)$, where $f^*$ is the global optima. Hence, the regret bound is $r_h \leq \delta(h_n^*)$. To find a lower bound for the quantity $h_n^*$, we first relate $h(n)$ to Lemma 3 by

$$n > \sum_{\tau=1}^{C\sum_{l=0}^{h(n)+\Xi_n-1}\delta(l)^{-d}} \rho_\tau \geq \sum_{\tau=1}^{\sum_{l=0}^{h(n)+\Xi_n-1}|I_l|} \rho_\tau,$$

where the first inequality comes from the definition of $h(n)$, and the second follows from Lemma 2. Then, from Lemma 3, we have $h_n^* \geq h(n) - 1$. Therefore, $r_n \leq \delta(h_n^*) \leq \delta(h(n) - 1)$. $\qquad \square$

**Assumption 5.** (Decreasing diameter revisit) The decreasing diameter defined in Assumption 3 can be written as $\delta(h) = c_1 \gamma^{h/D}$ for some $c_1 > 0$ and $\gamma < 1$ with a division procedure that requires $c_2$ function evaluations per node expansion.

**Corollary 1.** Assume Assumptions 1, 3, 4, and 5. Then, if $d = 0$, with probability at least $1 - \eta$,

$$r_N \leq O\left(\exp\left(-\frac{N + N_{gp}}{c_2 C D \bar{\rho}_t}\right)\right).$$

If $d > 0$, with probability at least $1 - \eta$,

$$r_N \leq O\left(\left(\frac{1}{N + N_{gp}}\right)^{1/d}\left(-\frac{c_2 C \bar{\rho}_t}{1 - \gamma^{d/D}}\right)^{1/d} \gamma^{-\frac{1}{D}}\right).$$

*Proof.* For the case $d = 0$, we have $n \leq \sum_{\tau=1}^{C\sum_{l=0}^{h(n)+\Xi_n}\delta(l)^{-d}}\rho_\tau \leq \sum_{}^{C(h(n)+\Xi_n+1)}\bar{\rho}_t$, where the first inequality follows from the definition of $h(n)$, and the second comes from the definition of $\bar{\rho}_t$ and the assumption $d = 0$. The second inequality holds for $\bar{\rho}_t$ that only considers $\rho_\tau$ with $\tau \leq t$. This is computable, because $\tau \leq t$ by construction. Indeed, the condition of Lemma 3 implies $t \geq \sum_{l=0}^{h'+\Xi_n}|I_l|$. Therefore, the two inequalities hold, and we can deduce that $h(n) \geq \frac{n}{C\bar{\rho}_t} - \Xi_n - 1$ by algebraic manipulation. By Assumption 5, $n = (N + N_{gp})/c_2$. With this, substituting the lower bound of $h(n)$ into the statement of Theorem 2 with Assumption 5,

$$r_N \leq c_1 \exp\left(-\left[\frac{N + N_{gp}}{c_2 D}\frac{1}{C\bar{\rho}_t} - \Xi_n - 2\right]\ln\frac{1}{\gamma}\right).$$

Similarly, for the case $d > 0$,

$$n \leq \sum_{\tau=1}^{C \sum_{l=0}^{h(n)+\Xi_n} \delta(l)^{-d}} \rho_\tau \leq \sum_{\tau=1}^{c^{-d} C \frac{\gamma^{-(h(n)+\Xi_n+1)d/D}-1}{\gamma^{-d/D}-1}} \bar{\rho}_t,$$

and hence $c\gamma^{\frac{h(n)+\Xi_n}{D}} \leq \left(\frac{n(1-\gamma^{d/D})}{C\bar{\rho}_t}\right)^{-1/d}$ by algebraic manipulation. Substituting this into the result of Theorem 2, we arrive at the desired result. $\qquad\square$

## B  With a Concrete Division Procedure

In this section, we prove the main result in the paper. In Theorem 1, we show that the exponential convergence rate bound $O\left(\lambda^{N+N_{gp}}\right)$ with $\lambda < 1$ is achieved *without* Assumptions 3, 4 and 5 and *without* the assumption that $d = 0$.

**Theorem 1.** Assume Assumptions 1 and 2. Let $\beta = \sup_{x,x'\in\Omega} \frac{1}{2}\|x-x'\|_\infty$. Let $\lambda = 3^{-\frac{\alpha}{2C\bar{\rho}_t D}} < 1$. Then, *without* Assumptions 3, 4 and 5 and *without* the assumption on $d$, with probability at least $1 - \eta$, the regret of IMGPO with the division procedure in Algorithm 1 is bounded as

$$r_N \leq L(3\beta D^{1/p})^\alpha \exp\left(-\alpha\left[\frac{N+N_{gp}}{2C\bar{\rho}_t D} - \Xi_n - 2\right]\ln 3\right) = O\left(\lambda^{N+N_{gp}}\right).$$

*Proof.* To prove the statement, we show that Assumptions 3, 4, and 5 can all be satisfied while maintaining $d = 0$. From Assumption 2, and based on the division procedure that the algorithm uses, $\sup_{x\in\omega_{h,i}} \ell(x, c_{h,i}) \leq \sup_{x\in\omega_{h,i}} L\|x - c_{h,i}\|_p^\alpha \leq L\left(3^{-\lfloor h/D\rfloor}\beta D^{1/p}\right)^\alpha$. This upper bound corresponds to the diagonal length of each hyperrectangle with respect to $p$-norm, where $3^{-\lfloor h/D\rfloor}\beta$ corresponds to the length of the longest side. We fix the form of $\delta$ as $\delta(h) = L3^\alpha D^{\alpha/p}3^{-h\alpha/D}\beta^\alpha \geq L(3^{-\lfloor h/D\rfloor}\beta D^{1/p})^\alpha$, which satisfies Assumption 3. This form of $\delta(h)$ also satisfies Assumption 5 with $\gamma = 3^{-\alpha}$ and $c_1 = L3^\alpha D^{\alpha/p}\beta^\alpha$. Every hyperrectangle contains at least one $\ell$-ball with a radius corresponding to the length of the shortest side of the hyperrectangle. Thus, we have at least one $\ell$-ball of radius $\nu\delta(h) = L3^{-\alpha\lceil h/D\rceil} \geq L3^{-\alpha}3^{-\alpha h/D}$ for every hyperrectangle with $\nu \geq 3^{-2\alpha}D^{-\alpha/p}$. This satisfies Assumption 4. Finally, to show $d = 0$ in this case, we note that, by Assumption 2, the volume $V$ of an $\ell$-ball of radius $\nu\delta(h)$ is proportional to $(\nu\delta(h))^D$ as $V_D^p(\nu\delta(h)) = (2\nu\delta(h)\Gamma(1+1/p))^D/\Gamma(1+D/p)$. Now, by definition, the $\delta(h)$-optimal space $X_{\delta(h)}$ is covered by an $\ell$-ball of radius $\delta(h)$, and is therefore covered by $(\delta(h)/(\nu\delta(h)))^D = \nu^{-D}$ $\ell$-balls of radius $\nu\delta(h)$. Therefore, the number of $\ell$-balls does not depend on $\delta(h)$ in this case, which means $d = 0$. Now that we have satisfied Assumptions 3, 4, and 5 with $d = 0$, $\gamma = 3^{-\alpha}$, and $c_1 = L3^\alpha D^{\alpha/p}\beta^\alpha$, we follow the proof of Corollary 1 and deduce the desired statement. $\qquad\square$

## Footnotes

[1]Thus, the probability in this analysis should be seen as that of *the subjective view*. If we assume that $f$ is indeed a sample from the GP, we have the same result with *the objective view* of probability.

## References for Supplementary Material

[1] N. Srinivas, A. Krause, M. Seeger, and S. M. Kakade. Gaussian Process Optimization in the Bandit Setting: No Regret and Experimental Design. In *Proceedings of the 27th International Conference on Machine Learning (ICML)*, pages 1015–1022, 2010.

[2] R. Munos. Optimistic optimization of deterministic functions without the knowledge of its smoothness. In *Proceedings of Advances in neural information processing systems (NIPS)*, 2011.