[Reviews · NeurIPS 2015]

Submitted by Assigned_Reviewer_1

The paper presents a new algorithm for global optimization that avoids delta-cover sampling and that achieves exponential regret, generalizing this way the work of Freitas el al., 2012 that relies on an impractical sampling procedure.

The paper is well written and is easy to follow and the role of the new algorithm within the current literature of bounded-based search methods is well explained. In this sense, I think that this work deserves some attention from the community.

I vote 5, however, a not a better score because I think that the experimental section is below the standard of what one expect to see in a BO paper. My main critics to this section are:

- The authors do no compare the methods for multiple initial evaluations of *f* as it is the standard in BO methods. Therefore no meaningful statical comparison of the methods can be obtained.

- Although they propose a bounded-based search method, the authors should compare they results with state of the art of BO approaches. Information theoretic approaches, such as, Entropy Search, are not used in the experiments.

- All the experiments are carried out in synthetic functions (whose maximum dimension is 6). No real wetlab of parameter tuning experiment is used to illustrate the performance of the method in real scenarios.

Summary: The paper presents a new global optimization algorithm that avoids delta-cover sampling and that achieves exponential regret. The paper is nice, with some interesting theoretical results but in my opinion the experimental section is below the standards of NIPS paper.

Submitted by Assigned_Reviewer_2

The paper presents a variant of Bayesian Global Optimization where future samples are generated through a space/interval partitioning algorithm.

Using UCB, this avoids the (internal) global optimization problem of how to choose the next sample.

Theoretical analysis shows that the approach yields theoretical improvements (exponential regret), thereby improving over previous work.

The algorithm is clearly described and explained/illustrated with an example.

The paper makes a novel contribution and is overall clearly presented.

I have one reservation with this paper.

The experimental results seem to be only for 1D test functions (not explicitly stated).

I suspect that the interval partitioning approach does not scale well to higher dimensions (hyperrectangles) because the "resolution" would be required to grow exponentially.

The approach is related to the well-known DIRECT algorithm, which is known to suffer badly when the dimensionality of the problem increases.

I think something at least needs to be said about this in the paper.

It does not change the theoretical contribution but is clearly significant for any practical purposes.

Minor comments:

p.2, UCB is considered "for brevity".

Does this mean you could do something with expected improvement, for example?

I got the feeling it had to be UCB.

p.3 "...we simultaneously conduct global and local searches based on all the candidates of the bounds."

I couldn't understand this statement.

p.4 "At n=16, the far right...but no function evaluation occurs."

Can you say why for clarity?
Summary: The paper presents a variant of Bayesian Global Optimization where future samples are generated through a space/interval partitioning algorithm, which yields theoretical improvements (exponential regret).

The work appears sound and novel but seems to be only evaluated on 1D test problems.

Submitted by Assigned_Reviewer_3

The paper describes a method of Bayesian global optimization which does not require d-cover sampling or auxiliary optimization.

The paper is organized such that the main contributions of the paper are well described in the main section.

The authors provide a good illustrative example of the algorithm in section 3.2, which gives the reader a nice high level understanding.

Section 3.3 also clearly communicates the workings of the algorithm programmatically for future implementation/replication.

Experimental results as well prove to be quite exciting.

Additionally, the authors note that the paper brings together ideas from the Bayesian global optimization literature that do and do not use continuity estimation.

Summary: The paper describes Infinite-Metric GP Optimization (IMGPO), an algorithm for Bayesian optimization without the need for auxiliary optimization or d-cover sampling.

The paper is not only clearly written, but provides a strong contribution to the NIPS community.

Submitted by Assigned_Reviewer_4

(light review) Abstract formulation is redondant l49 the function l72 misplaced parenthesis
Summary: Being unfamiliar with GP and global optimization, the reviewer's evaluation is an educated guess.

The paper is well written and rather didactic, making good use of graphs to explain the procedures. Both theoretical and experimental results seem to conclusively demonstrate the advantage of the authors' approach.

Author Feedback
Author rebuttal: == Reviewer 1 ==
Thank you for the comments. Please note that we did compare our algorithm with the most closely related state-of-the-art algorithms (SOO and BaMSOO). Also, we used standard benchmark test functions taken from the global optimization literature.

Thank you for pointing out that many other experimental results in Bayesian optimization are reported with random initial evaluations. Based on that suggestion, we re-ran our experiments and found that, under those conditions, our algorithm actually performs better compared to the state-of-the-art methods.

For evaluation on a non-synthetic function, we are now successfully using our algorithm to optimize the parameters of a filtering algorithm (based on RKHS embedding of probability densities). We would be able to report these results in a final version of the paper.

== Reviewer 2 ==
Thank you for the comments. Please note that we evaluated our algorithm with 1D - 6D test functions (not just 1D test functions). Because the dimensionality of each test function was indicated only in the figure title, it could have been confusing, so we have now added that information in the text as well.

For the concern regarding the input dimensionality, please see the last section of this feedback, titled ``== Input Dimensionality ==''.

== Reviewer 4 ==
Thank you for the comments. We have revised the results by showing the average performance as suggested. The relative performances across the algorithms stayed the same. Indeed, it produced more preferable results for our algorithm.
We agree with the comment on the log regret in Figure 2. Ideally, we should show the results in both scales, but we followed the convention that has been used in previous work.

For the concern regarding the input dimensionality, please see the last section of this feedback, titled ``== Input Dimensionality ==''.

For the real parameter tuning, we are now successfully using our algorithm to optimize the parameters of a filtering algorithm (based on RKHS embedding of probability).

== Reviewer 5 ==
Thank you for the comments.

== Reviewer 6 ==
Thank you for the comments. We revised the paper accordingly.

== Input Dimensionality ==
As reviewer 2 suggests (and reviewers 1 and 3 imply), scaling up to high dimensions is a key issue in Bayesian optimization in general. However, unlike DIRECT type algorithms, our algorithm is as good (bad) as GP-UCB in terms of scaling up to higher dimensions. While the required ``resolution'' of the partitioning may suffer badly from the curse of dimensionality, the required number of the function evaluations grows similarly to GP-UCB. This is because while refining the ``resolution'' of the partitioning, the function evaluations are avoided due to the UCB computed by GP. Notice that a typical BO algorithm has more serious problems in terms of the resolution required without the function evaluations in auxiliary optimization.

The dimensionality of the test functions used in this paper does not directly reflect limitations of our approach. We simply tested our algorithm with convenient standard benchmark functions in the related global optimization literature. It would be interesting to see how it works with moderate dimensionality. For higher dimensions, if we can assume ``effective dimensionality'', our approach could be applied by simply using random projection (Wang et al., 2013: Bayesian Optimization in High Dimensions via Random Embeddings) or learn it with Bayesian inference. A more ambitious approach would be to use a partitioning strategy that suffers much less from the curse of dimensionality.

We would add a refined version of the above discussion regarding input dimensionality to a final paper version, subject to overall space constraints.